# Comparing the Accuracy of Soil Moisture Estimates Derived from Bulk and Energy-Resolved Gamma Radiation Measurements

**DOI:** 10.3390/s25144453

**Published:** 2025-07-17

**Authors:** Sonia Akter, Johan Alexander Huisman, Heye Reemt Bogena

**Affiliations:** Agrosphere Institute (IBG-3), Forschungszentrum Jülich GmbH, 52428 Jülich, Germany; s.huisman@fz-juelich.de (J.A.H.); h.bogena@fz-juelich.de (H.R.B.)

**Keywords:** soil moisture monitoring, gamma radiation, confounding factors, wavelet transform coherence

## Abstract

Monitoring soil moisture (SM) using permanently installed gamma radiation (GR) detectors is a promising non-invasive method based on the inverse relationship between SM and soil-emitted GR. In a previous study, we successfully estimated SM from environmental gamma radiation (EGR) measured by a low-cost counter-tube detector. Since this detector type provides a bulk GR response across a wide energy range, EGR signals are influenced by several confounding factors, e.g., soil radon emanation, biomass. To what extent these confounding factors deteriorate the accuracy of SM estimates obtained from EGR is not fully understood. Therefore, the aim of this study was to compare the accuracy of SM estimates from EGR with those from reference ^40^K GR (1460 keV) measurements which are much less influenced by these factors. For this, a Geiger–Mueller counter (G–M), which is commonly used for EGR monitoring, and a gamma spectrometer were installed side by side in an agricultural field equipped with in situ sensors to measure reference SM and a meteorological station. The EGR_G–M_ and spectrometry-based ^40^K measurements were related to reference SM using a functional relationship derived from theory. We found that daily SM can be predicted with an RMSE of 3.39 vol. % from ^40^K using the theoretical value of *α* = 1.11 obtained from the effective ratio of GR mass attenuation coefficients for the water and solid phase. A lower accuracy was achieved for the EGR_G–M_ measurements (RMSE = 6.90 vol. %). Wavelet coherence analysis revealed that the EGR_G–M_ measurements were influenced by radon-induced noise in winter. Additionally, biomass shielding had a stronger impact on EGR_G–M_ than on ^40^K GR estimates of SM during summer. In summary, our study provides a better understanding on the lower prediction accuracy of EGR_G–M_ and suggests that correcting for biomass can improve SM estimation from the bulk EGR data of operational radioactivity monitoring networks.

## 1. Introduction

Soil moisture (SM) is a key component of the terrestrial ecosystem, influencing hydrological, ecological, and atmospheric processes [1]. It has a direct impact on local, regional, and eventually global hydrological cycles by controlling water infiltration, storage, and movement within the soil, which in turn determines how rainfall is distributed between groundwater recharge and surface runoff [2,3,4]. Additionally, SM influences ecological processes by regulating the complex interactions among water resources, energy fluxes, and biogeochemical cycles [5]. Through its influence on evapotranspiration, SM plays an essential role in the exchange of energy between the land surface and atmosphere, thereby affecting weather patterns and climate dynamics [6]. Hence, estimating the spatial and temporal dynamics of soil moisture accurately is crucial. Several measurement techniques such as gravimetric sampling, geophysical measurements, sensor networks, and air-borne and space-borne remote sensing have been developed to estimate SM over a broad range of scales [1,7].

Measuring gamma radiation (GR) originating from the decay of natural soil radionuclides, such as ^40^K, ^232^Th, and ^238^U, is an emerging non-invasive technique for monitoring the temporal variability of SM within a relatively large footprint (~100′s m^2^) [8,9]. The measurement principle relies on the inverse relationship between SM and GR due to the increased attenuation of GR emitted from soil with increasing SM content [10]. Airborne GR measurements have been used to estimate spatial variation in SM content [11,12]. However, there are several challenges associated with obtaining SM estimates from airborne GR measurements, such as the need to correct for variability in atmospheric vapor and density [13]. In addition, the spatial distribution of soil radionuclides that determine the background radiation is often not known [7]. Therefore, permanently installed ground-based GR measurements are more promising than air-borne GR measurements in terms of measurement accuracy and operational costs for monitoring field-scale SM dynamics [7].

Measurements of GR with permanently installed sensors have already been used to monitor SM [8,14,15]. It was found that the GR signal emitted by ^40^K is a more reliable source for SM estimation compared to other soil radionuclides, specifically ^238^U [8,9]. This is because ^40^K decays by emitting gamma radiation at a single energy of 1460 keV and the confounding factors (e.g., atmospheric conditions) have less influence on such high-energy GR [9]. In contrast, ^238^U has a decay chain consisting of several radionuclides before ending as a stable isotope, emitting 458 gamma rays with energies ranging between 20 and 3300 keV throughout the process [13]. Moreover, some radionuclides (intermediate or stable) formed during the decay of ^238^U can be influenced by atmospheric conditions. As an example, radon (^222^Rn), a gaseous progeny of the ^238^U decay series, can escape from the soil into the atmosphere and decay further into the stable radionuclides ^214^Bi and ^214^Pb [13]. As a result, estimating SM from ^238^U GR intensity may be less accurate due to the additional atmospheric GR contribution [9,11].

Gamma-ray spectrometers are widely used to measure GR emitted by natural soil radionuclides [8,12,14,15]. The spectrometers contain scintillator material based on NaI (Tl), CsI, or LaBr_3_ (Ce) crystals, which absorb gamma photons and emit light proportional to the energy of absorbed photons. Therefore, spectrometers can provide information on both the GR intensity and the energy of incident GR. By integrating energy-resolved measurements, spectrometers can also provide information on the bulk environmental gamma radiation (EGR) intensity. In addition, GR spectrometers are sensitive to low-energy gamma rays, which facilitates the detection of extremely low concentrations of artificial nuclides [16]. However, spectrometric GR detectors are expensive and require maintenance for long-term operation [16,17].

Geiger–Mueller (G–M) counters are a type of gaseous ionization GR detectors widely used in EGR monitoring networks that have been established to provide early warning for nuclear accidents [17,18,19,20,21]. Because these detectors are relatively cheap and robust, they can be successfully operated for many years [16,17]. However, a major limitation of G–M counters is that they only provide EGR measurements integrated over a wide energy range (~0–8000 keV). Therefore, it is not possible to extract the GR intensity of a specific radionuclide in a particular energy window (e.g., ^40^K GR intensity at 1460 keV) from the EGR measurements. This limitation is linked to the measurement principle of G–M counters, which use a low-pressure inert gas-filled tube. When ionized by incident radiation, the gas produces an output pulse that does not provide information about the energy of the incident radiation [16]. Due to the different measurement principles, G–M counters have a different energy sensitivity compared to scintillator-based spectrometers [16].

It is well known that EGR measurements are influenced by several confounding factors since the signatures of all soil radionuclides are embedded in it [22,23]. As an example, clear spikes in EGR time series were observed during and up to three hours after precipitation [22,24]. This is related to rain washout of radon progenies leading to an abrupt increase in EGR measurements [25]. Apart from atmospheric washout, EGR measurements can also be affected by radon variability due to variable mixing and exchange processes in the atmospheric boundary layer [26]. In addition to the contribution of GR emitted by terrestrial radionuclides, secondary cosmic radiation (SCR) associated with the interaction of primary cosmic radiation with the Earth’s atmosphere also significantly contributes to the EGR measurements [22,24]. Variations in SCR can occur due to changes in atmospheric pressure and incoming neutron intensity leading to fluctuations in EGR measurements over time [23,24]. Another confounding factor, biomass, can increase inaccuracies in SM estimates from EGR measurements by further attenuating the GR signal [15,24,27].

In a previous study, we obtained SM estimates from EGR measured by proportional counter tube-based gaseous ionization GR detectors with a similar measurement principle as G–M detectors [24]. The accuracy of the obtained SM estimates was 7–9 vol.%, which is lower than for other in situ measurement techniques. Despite this relatively low accuracy, the long-term availability of ERG data in Europe through the EURDEP network motivates us to carry out further investigations to improve the interpretation of EGR measurements and to facilitate SM analysis at the continental-scale using EURDEP data. In our previous study, we already applied correction methods to account for atmospheric radon washout and SCR variability, but the influence of other confounding factors such as soil or atmospheric radon not associated with precipitation, biomass, and detector-specific characteristics (e.g., energy sensitivity) was not yet investigated. In addition, it was difficult to understand how the confounding factors affected the EGR measurements in our previous experimental set-up, and to what extent these factors reduced the accuracy of SM estimates derived from EGR measurements.

Within this context, the aim of the present study was to compare the accuracy of SM estimates derived from low-cost G–M-based EGR (EGR_G–M_) and spectrometry-based reference ^40^K GR (1460 keV) measurements, which are less influenced by the confounding factors. To better understand the influence of confounding factors on EGR_G–M_ measurements, we used additional spectral GR information and in situ measurements for data interpretation. The remainder of this paper is organized as follows. First, we investigated the correlation of EGR_G–M_ measurements with spectrometry-based EGR (EGR_S_) measurements, which are comparable to EGR_G–M_ measurements, over different time scales to understand the detector-specific characteristics of G–M counters such as energy sensitivity. Next, we investigated the correlation of EGR_S_ with ^214^Bi (which is an indicator of radon variability that is driven by atmospheric processes but not related to atmospheric washout due to precipitation) and ^40^K to understand the variability of EGR caused by those radionuclides over different time scales. In addition, we also investigated the correlation of SM with all GR measurements. For analyzing the correlation between two time series, we used wavelet analysis technique as it is a multi-resolution technique to examine localized variations in time-series data within the time-frequency domain [28,29]. In a final step, we estimated SM from EGR_G–M_ and ^40^K GR measurements based on the functional relationship between reference SM and GR derived from GR attenuation theory. Here, we also identified the results for the period when biomass was present to assess the influence of biomass on EGR_G–M_ and ^40^K GR signals. Finally, we compared the results of EGR_G–M_ with reference ^40^K GR to evaluate how confounding factors reduce the accuracy of SM estimates obtained from EGR_G–M_ measurements.

## 2. Materials and Methods

### 2.1. Experimental Site

The experiment was conducted at the agricultural research station Selhausen (50.865° N, 6.447° E, 101 m above sea level) in western Germany. The station is part of the TERENO (TERrestrial ENvironmental Observatories; [30]) and ICOS (Integrated Carbon Observation System; [31]) research networks and falls within climate class Cfb (temperate maritime climate zone). The mean annual temperature is 10.2 °C and the mean annual precipitation is 714 mm. The soil is characterized by a silty loam texture with 1% organic carbon content. More detailed information on this site can be found in Bogena et al. [32].

This study considered an observation period from February 2023 to March 2024 in which winter wheat was the main crop with a growing season from October 2022 to July 2023. Due to a technical problem, the harvest of winter wheat adjacent to the measurement footprint of the GR detectors was delayed and a late harvest was eventually conducted in August 2023. After the harvest of winter wheat, a mix of cover crops was planted in September 2023 and harvested at the end of January 2024. The harvesting process ensured minimal soil disturbance as plant residues were left during harvesting. However, the soil was disturbed once in September 2023, when it was ploughed before planting cover crops. Plant height was measured at regular intervals of approximately 15–20 days throughout the entire investigation period. A time window with high fresh biomass and dry biomass (after ripening) was identified based on available crop information (i.e., growth stages, plant height) to investigate the importance of the shielding effect of biomass.

### 2.2. Experimental Setup

The experimental setup consisted of a spectrometric GR detector (SARA, ENVINET GmbH, Haar, Germany), a G–M detector (MIRA, ENVINET GmbH, Haar, Germany), five reference SoilNet soil moisture stations [33], and an eddy covariance (EC) station with meteorological sensors. The SARA detector uses scintillation material based on NaI crystals to measure energy-resolved GR in the energy range of 30–3000 keV with an accuracy of ±10%. A two-step process was used to obtain radionuclide-specific GR intensity from the measured GR spectrum [34]. First, the flux rate (number of incident photons over the cross-sectional area of the detector) was calculated considering a specific energy window of the GR spectrum. Subsequently, the GR intensity was calculated from the flux rate using energy-dependent conversion coefficients. In the case of a radionuclide with several gamma lines, such as ^214^Bi, all lines throughout the measured spectrum (e.g., at 609, 1120, 1764, 2204 keV) were considered and summarized to obtain the GR intensity. In the same way, the bulk GR intensity or EGR intensity was calculated considering all the gamma lines of the spectrum emitted from all ambient radionuclides in the given energy range of the spectrometer. It should be noted that the SARA detector response is sensitive to temperature change and can drift due to the aging of detector components causing the same gamma-ray energy to appear at different positions (channels) in the spectrum over time. Therefore, continuous and automatic energy stabilization is carried out for SARA in real time using an active energy-stabilization algorithm which employs a temperature sensor and a naturally occurring reference radionuclide such as ^40^K (1460.8 keV). Hence, the GR response factor stays relatively stable in the energy range of about 700–3000 keV but shows significant variation for low-energy GR below 500 keV (see the relative efficiency as a function of gamma-ray energy in Figure A1a of Appendix A).

The MIRA detector contains G–M counter tubes filled with inert gas at a low pressure to which a high voltage is applied. The counter tubes detect bulk EGR using the ionization effect, generating an output pulse for each detected ionization event regardless of the gamma-ray energy, which is then counted. Two separate G–M counting tubes for low- and high-GR-intensity measurements are incorporated in the MIRA detector to facilitate a wide detection range of up to >10 Sv h^−1^. In this study, MIRA was used to measure EGR intensity with a 10 min integration time and a temperature-dependent intrinsic background compensation that is valid from −25 °C to 65 °C. The MIRA detector does not need continuous energy calibration as the SARA detector does. Instead, the functional calibration is based on the count rate response using a known test source (^137^Cs at 662 keV) in a fixed geometry. Therefore, MIRA gives a linear response in the 30–1000 keV range with a significant increase in sensitivity at energies above 1000 keV (Figure A1b). As both GR detectors were installed at a height of 1 m above ground, it can be assumed that 95% of the detected GR signals originated from an area with an approximate 7–15 m radius and 12–25 cm depth depending on GR energy [8,24].

The reference in situ SM was measured with SoilNet at five locations with a temporal resolution of 15 min (Figure 1). Each of the SoilNet stations was equipped with eight SMT100 sensors (Truebner GmbH, Neustadt, Germany). The sensors were installed in pairs at depths of 2, 5, 10, and 20 cm and were assumed to represent the following depth ranges: 0–3.5, 3.5–7.5, 7.5–15, and 15–25 cm, respectively. Additionally, the EC station was also equipped with two additional SM sensors (CS616, Campbell Scientific Inc., Logan, UT, USA), installed at a 2 cm depth parallel to each other representing SM measurements at a 0–3.5 cm depth. The SM measured at different depths by all stations was averaged horizontally to derive an average SM representative for each depth range. Since the vertical distribution of the sensors was not uniform, the mean reference SM (*θ_ref_*), for the top 25 cm soil within the measurement footprint of the GR detectors was obtained as follows:(1)$$\theta_{ref}=\frac{{(SM}_{1}\times d_{1}+{SM}_{2}\times d_{2}+{SM}_{3}\times d_{3}+{\;SM}_{4}\times {\;d}_{4})}{d_{1}+{\;d}_{2}+d_{3}+d_{4}}$$
where *SM*_1_, *SM*_2_, *SM*_3_, and *SM*_4_ are the average SM values over the depth intervals 0–3.5 cm, 3.5–7.5 cm, 7.5–15 cm, and 15–25 cm, respectively, and the variables *d*_1_, *d*_2_, *d*_3_, and *d*_4_ represent the thickness of the corresponding soil layers in centimeters.

### 2.3. Data Filtering, Correction and Gap Filling

The EGR and ^214^Bi measurements require filtering for the washout of atmospheric radon progenies during and three hours after precipitation [24]. The precipitation time series throughout the whole investigation period was obtained from a nearby climate station ~400 m northeast of the experimental site. Following Akter et al. [24], periods with precipitation and up to three hours after precipitation were identified and excluded from further analysis. No filter was used for ^40^K GR measurements as these measurements did not show spikes in radiation associated with precipitation. The EGR measurements also require correction for the contribution of SCR [24]. The long-term fraction of SCR in EGR is assumed to be constant for the experimental site. However, atmospheric pressure and incoming neutron intensity can cause short-term variations in SCR leading to variations in EGR measurements [24]. Hence, for the purpose of this study, it is sufficient to correct the EGR measurements for short-term SCR variability. The atmospheric pressure data were obtained from the EC station situated next to the GR detectors. Incoming neutron intensity data were obtained from the neutron monitor database (NMDB https://www.nmdb.eu/nest/ (accessed on 12 October 2024)) for the closest station at the Jungfraujoch (JUNG, Switzerland). We used the two time series to investigate the influence of short-term SCR variability on the EGR measurements and corrected accordingly following Akter et al. [24] without including the constant contributions of muon and neutron flux to the SCR.

The GR data were measured continuously without any gap throughout the whole investigation period. However, data filtering for the removal of radon washout resulted in a few short-time data gaps and those gaps were filled by linear interpolation. Four of the SoilNet stations were removed at the end of July 2023 to enable crop harvest and reinstalled at the beginning of September 2023 after ploughing and planting cover crops. The resulting one-month data gap was filled using the SM time series measured by the remaining SoilNet station installed near the GR detectors. To do so, a linear regression relationship was established between the mean SM and the SM measurements from that single station for each depth range. Since the plant height was measured biweekly, a linear interpolation method was used to estimate daily plant height data.

### 2.4. Wavelet Analysis

Wavelet analysis is a multi-resolution technique used to explore localized variations in a time series with a focus on the time-frequency space. It breaks down time-series data into different time-frequency scales, allowing the signal to be represented as scaled and shifted versions of the original wavelet [29]. By doing so, it enables the detailed examination of changes in correlation and lag times between variables across different time scales and identifies localized, intermittent periodic patterns [28]. Unlike traditional mathematical approaches, such as Fourier Analysis, which assume processes are time-invariant, wavelet analysis provides a more flexible approach by handling non-stationary data [35]. More information on the background of wavelet analysis can be found in Torrence and Compo [29], Grinsted et al. [28], and Weigand et al. [35].

Wavelet transforms can be divided into two classes: continuous wavelet transforms (CWT) and discrete wavelet transforms (DWT). In this study, CWT was selected because of its ability to extract features while effectively managing complex and noisy time-series data [28,35]. We used the Morlet wavelet that is widely used in CWT [35,36] because it provides a good balance between time and frequency localization [28]. As part of the wavelet analysis, the wavelet transform coherence (WTC) was obtained. It is a localized correlation coefficient that reveals not only when two CWTs are correlated, but also at which frequencies. It allows for a more nuanced understanding of interactions over varying timescales [28]. The WTC is often presented in a time-frequency plot called a scalogram, which shows time on the horizontal axis and frequency (or scale) on the vertical axis. These frequencies represent different time scales, with lower frequencies corresponding to longer-term trends (e.g., seasonal or annual trends), and higher frequencies reflecting short-term variations (e.g., daily variations). The arrows in the scalogram represent the phase relationship between two time series over time and frequency. Arrows pointing to the right indicate that the two time series are in phase (positively correlated), while arrows pointing to the left indicate they are in anti-phase (negatively correlated). Arrows pointing straight up or down indicate a phase shift of −90° or +90°, respectively, while diagonal arrows represent intermediate phase differences between two time series. The color spectrum in the scalogram represents the wavelet power or the strength of coherence (correlation) between two time series at different times and frequencies, which can vary between 0 (no coherence) and 1. A value of 0.6 is often used to determine a scale decorrelation length for the Morlet wavelet [29]. The Cone of Influence (COI) in the scalogram marks the region (near the edges of the plot) where results could be influenced by edge effects causing distortion. Data outside the COI should be interpreted cautiously, as the results may be less reliable. The wavelet analysis was performed using the Cross Wavelet and Wavelet Coherence Toolbox in MATLAB R2024b (http://grinsted.github.io/wavelet-coherence/ (accessed on 27 July 2024)), which was developed by Grinsted et al. [28]. For more detailed information on how to interpret the results of a wavelet analysis, the reader is referred to Robinson et al. [36].

### 2.5. Soil Moisture Estimation

Both the ^40^K GR and EGR_G-M_ measurements were calibrated against the mean SM of the top 25 cm of soil. For this, a functional relationship derived from established GR attenuation theory assuming homogeneous soil was used [37,38]:(2)$$\frac{GR}{GR_{dry}}=\frac{1}{1+\alpha \frac{\theta \rho_{w}}{\rho_{b}}}$$
where *θ* is the SM in volumetric percentage (vol. %), *GR* is the measured ^40^K GR or EGR_G-M_ intensity, *GR_dry_* is the maximum ^40^K GR or EGR_G-M_ intensity when *θ* = 0 vol. %, *α* is the effective ratio of the GR mass attenuation coefficients for water and solid phases, *ρ_w_* is the density of water, and *ρ_b_* is the dry bulk density of the soil. For the present study, *ρ_b_* was measured from soil samples of 30 cm length and 5 cm diameter obtained using a HUMAX soil corer (Martin Bruch AG, Rothenburg, Switzerland). We collected three soil cores within the measurement footprint of the GR detectors. Each core was subdivided into six subsamples of 5 cm length and oven-dried at 105 °C for 24 h. The average *ρ_b_* of the top five samples of each core was used to calculate the mean *ρ_b_* of the top 25 cm of soil.

For the ^40^K GR measurements at 1460 keV, the mass attenuation coefficients for the solid and water phase are well known. Their ratio results in an α value of 1.11 [39], which is in agreement with the findings of Baldoncini et al. [27]. The α value of 1.11 is only valid for gamma rays within an energy range of 500–1500 keV [39]. Becker et al. [15] found an α value of 0.63 for ^40^K GR measurements while using a full spectrum analysis for SM estimation, suggesting that further research and measurements at additional sites are required to establish appropriate values for α. As was already shown in Akter et al. [24], the α-value of 1.11 is also not applicable for EGR_G-M_ measurements that cover a broader energy range. Therefore, we calibrated the *α* parameter only for the EGR_G-M_ measurements. More detailed information on calibrating EGR measurements can be found in Akter et al. [24]. The remaining unknown parameter, *GR_dry_*, was calibrated for both the ^40^K GR and EGR_G-M_ measurements. After calibration, SM was estimated from the ^40^K GR and EGR_G-M_ measurements using a reorganized form of Equation (2).

To establish a relationship between ^40^K GR and SM, Becker et al. [15] also accounted for additional site-specific hydrogen pools that attenuate GR signals such as lattice water (hydrogen structurally bound within the crystal lattice of clay minerals) and soil organic carbon (hydrogen physically or chemically bound to soil organic matter) in addition to pore water. In this study, detailed information on lattice water and soil organic carbon water was not available. However, lattice water and soil organic carbon water should be approximately constant over a year [40]. Therefore, we did not consider corrections for lattice water and soil organic carbon content here.

We also excluded the data collected during snow events and in the few days after reinstalling the reference SM sensors following harvest from the analysis to reduce the uncertainty of the SM estimates.

## 3. Results and Discussion

### 3.1. Analysis of Daily Gamma Radiation (GR) Time Series

The daily time series of precipitation, temperature, and GR measurements obtained from the GR detectors MIRA (EGR_G-M_) and SARA (EGR_S_; GR from ^40^K and ^214^Bi), referencing in situ SM, plant height, and land cover are shown in Figure 2. The measured GR data with a 10 min temporal resolution were noisy due to uncertainty in the count rate. Moreover, different diurnal trends were observed for different kinds of GR measurements, which also varied throughout the year (Figure A2 and Figure A3). As an example, an increase in ^40^K GR intensity was observed with an increase in temperature, peaking at midday during the summer (Figure A2). This temperature-dependent daily cycle could be a consequence of temperature-dependent SM changes during dry summer days. With the increase in temperature, SM decreases due to evaporation leading to an increase in ^40^K GR intensity during the day. In agreement with this finding, Amestoy et al. [11] also reported a pronounced diurnal cycle of ^40^K GR intensity possibly related to the diurnal cycle of SM. It is interesting to note that a small time-lag between the daily ^40^K GR intensity and temperature can occasionally be observed (Figure A2). This delay can be attributed to the depth-dependent changes in SM during evaporation, where the near-surface soil dries out quickly on dry summer days due to evaporation, but deeper parts of the soil dry out more slowly. In contrast, the increase in ^40^K GR intensity begins as soon as the topsoil layer starts to dry [11]. However, it should be noted that careful consideration is required when interpreting diurnal SM variations, as apparent diurnal SM variations may result from temperature effects on sensor electronics. Therefore, future studies with co-located spectrometers from different manufacturers are required to confirm that the observed changes in ^40^K GR intensity are related to actual changes in the soil.

On several cloud-free summer days, EGR measurements showed an increase during the night followed by a sharp decrease after sunrise (Figure A3). This is attributed to atmospheric temperature inversion layers that retain radon and radon progenies during the night which leads to an increase in EGR. After sunrise, surface warming destroys these inversion layers [26], which leads to a decrease in EGR [41,42]. To minimize the measurement uncertainties associated with short integration times (i.e., hourly), a daily mean value smoothed with a three-day rolling mean filter was used for GR measurements in the remainder of this study. Similarly, the reference SM and temperature measurements were averaged to create a daily time series.

Table 1 presents summary statistics of the daily GR time series. Although the coefficient of variation for EGR_G–M_ and EGR_S_ measurements were found to be similar, the mean EGR_G–M_ is higher than the mean EGR_S_ (difference of 18 nSv h^−1^), which is in agreement with the findings of Casanovas et al. [34]. The reason for this discrepancy could be that the G–M detectors generally respond more strongly to GR above >662 keV because they are calibrated with a ^137^Cs source [34]. For the radionuclide-specific GR measurements of SARA, the coefficient of variation was higher compared to the EGR measurements.

The EGR_G-M_ measurements were largely consistent with EGR_S_ measurements for most of the time (Figure 2c,d). A scatter plot between measurements of both detectors corroborates this consistency and results in a Pearson correlation coefficient of 0.91 (Figure A4). However, there are a few time windows where the dynamic variations in both types of measurements were different (highlighted in Figure 2c,d). Since the EGR measurements consist of gamma rays emitted from all radionuclides present in the environment, the contribution of SCR is also embedded in it [43]. Changes in atmospheric pressure and incoming neutron intensity lead to the short-term variation in SCR [23,24]. Therefore, it was expected that the EGR measurements would be negatively correlated with atmospheric pressure and positively correlated with incoming neutron intensity [23,24]. In this study, the EGR_G-M_ measurements showed a weak negative correlation with atmospheric pressure (r = −0.28) and no correlation with incoming neutron intensity (r = 0.09). The EGR_S_ measurements showed no correlation with atmospheric pressure and incoming neutron intensity. This finding aligns with the fact that SCR significantly contributes to EGR at low energies (<300 keV) [44] and high energies (≥3000 keV) [45]. The contribution of SCR is lower for energies between 300 and 3000 keV, where the spectrometric monitoring system operates. Therefore, the minor discrepancies between the two types of GR detectors highlighted in Figure 2 may be attributed to differences in SCR contribution to the EGR_G-M_ and EGR_S_ measurements. Based on this analysis, the EGR_G-M_ measurements were corrected for short-term SCR variability caused by atmospheric pressure and incoming neutron intensity and are referred to as EGR_G-M′_ from here on. The EGR_S_ measurements were not corrected.

### 3.2. Wavelet Coherence Analysis

The WTC plot in Figure 3a shows the correlation between EGR_G-M_ and EGR_S_ measurements for different time frequencies. The correlation between the two types of measurements is unclear at the hourly time scale due to the high uncertainty in the count rate of the hourly data. However, a distinct positive correlation between the time series was observed on a daily time scale, specifically during the summertime (i.e., red arrows between a period of 16–32 h from May 2023 to August 2023 in Figure 3a). Between November 2023 and March 2024, this correlation was barely observed. The time-series data of those two types of measurements in Figure 3b also confirm that the discrepancies between them were mainly observed during spring- and wintertime (March 2023 to May 2023; November 2023 to March 2024). The consistency of the EGR_G-M_ and EGR_S_ measurements improved after correction for short-term SCR variability (EGR_G-M′_), specifically during the springtime (March 2023 to May 2023, light gray shaded area in Figure 3d). The Pearson correlation coefficient with EGR_S_ also improved from 0.91 to 0.96 when daily EGR_G-M′_ measurements were used (Figure A4 and Figure A5). At the same time, it is also evident that some discrepancies remained, especially during wintertime (November 2023 to March 2024, Figure 3d). After the unwanted noise due to SCR was removed, the EGR_G-M′_ and EGR_S_ measurements showed similar variability in GR albeit with a different magnitude. Consequently, the correlation between the two types of measurement improved for longer time periods between November 2023 and March 2024 (Figure 3c).

In a next step, the different types of GR measurements obtained from MIRA (EGR_G–M′_) and SARA (EGR_S_; GR from ^40^K and ^214^Bi) detectors were used to assess their coherence with the reference SM using WTC analysis (Figure 4). As expected, an anticorrelation between reference SM and different types of GR measurements was observed. In the WTC plots, the anticorrelation was not very distinct at the daily time scale but became more evident at the weekly or biweekly time scales (128 to 512 h). The GR measurements from ^40^K showed the highest correlation with SM (Figure 4c), while the lowest correlation was obtained for ^214^Bi (Figure 4d). The coherence spectra for ^40^K indicated a weak correlation between GR measurements from ^40^K and SM for the period between October 2023 and March 2024. The SM did not vary much during that period due to low evapotranspiration and the excess supply of precipitation (Figure 2a,b,g). This resulted in a weak correlation between SM and GR measurements from ^40^K in this period. A similar correlation trend over time was observed for EGR_S_ measurements, although the coherence of SM with EGR_S_ measurements was weaker than that with ^40^K GR measurements (Figure 4a,c). No distinct differences were visible in the correlation of SM with EGR_S_ and EGR_G–M′_ measurements (Figure 4a,b).

The coherence between different types of GR measurements obtained from the SARA detector was investigated to understand the influence of contributory radionuclides on EGR over time, and ultimately to better interpret the dependency of EGR on SM variations (Figure 5). In general, EGR_S_ showed a strong positive correlation with ^40^K and ^214^Bi GR measurements, particularly on the weekly time scale (128 to 256 h, Figure 5a,c). However, the coherence varied considerably with time. As an example, ^40^K GR intensity showed strong correlation with EGR_S_ during summertime and no correlation in wintertime on the weekly time scale, specifically from November 2023 to March 2024. As discussed earlier, the variability in SM during wintertime was limited. This likely led to minimal variation in ^40^K GR measurements, which in turn may have contributed to the lack of correlation between EGR_S_ and ^40^K during that period. Interestingly, ^214^Bi GR intensity showed stronger correlation with EGR_S_ during wintertime, which is also apparent in the daily time series (Figure 5c). This clearly suggests that ^214^Bi is an important contributor for EGR variability during the wintertime (February to March 2023 and November 2023 to March 2024). Given that ^214^Bi is a well-known radon progeny, the observed difference seems to be related to how the near-surface radon concentration was affected by complex soil and atmospheric processes. Although the time series were filtered for atmospheric radon-washout during and three hours after rain, it is well known that the near-surface radon concentration is higher and more variable in wintertime [26,46]. This is related to the reduced vertical mixing of radon in cold and cloudy daytime atmospheric boundary layers [26]. In addition, radon can be trapped in the soil during snow and ground frost events due to a substantial reduction in the diffusion coefficient for frozen soils [47]. Consequently, it may lead to increased radon exhalation rates from the soil if the temperature suddenly rises after snow or ground frost melting [48]. In agreement with this, ^214^Bi GR intensity for the current study increased sharply following a sudden rise in air temperature, which occurred after a longer period of temperatures below 0 °C (see the time windows right after the ground frost event from 26 February to 9 March 2023 and the snow event from 11–22 January 2024 in Figure 2b,f). Apart from snow or ground frost events, the soil is generally wet during wintertime (November 2023 to March 2024 in Figure 2f). Therefore, radon exhalation from the soil may have been limited after rainfall due to reduced diffusion [49]. As a result, radon exhalation from the soil could occur with a delay, leading to greater variability in EGR during wintertime.

### 3.3. Influence of Confounding Factors on Soil Moisture Estimation

To investigate the influence of confounding factors on SM estimates from EGR_G–M′_ measurements, we calibrated both the EGR_G–M′_ and reference ^40^K GR (peak intensity at 1460 keV) measurements against reference SM using Equation (2) (Figure 6). The mean *ρ_b_* for the top 25 cm soil was determined to be 1.2 g cm^−3^. Using the *α*-value of 1.11 derived from GR attenuation theory, we achieved a good fit for ^40^K GR using a value of 12.90 nSv h^−1^ for the parameter *GR_dry_*. The fitted relationship was used to predict SM from ^40^K GR measurements, which resulted in an RMSE of 3.39 vol. %. Becker et al. [15] reported an RMSE of 4.2 vol. % for estimating SM from ^40^K full spectral measurements using a fitted *α*-value of 0.63. The resulting RMSE values are on par with those reported for other field-scale soil moisture estimation methods, such as cosmic ray neutron probes with an RMSE of approximately 3 vol. % [40] and electromagnetic induction with an RMSE of 2–5 vol. % [50,51].

As shown in Akter et al. [24], the variable *α* needs to be estimated along with the variable *GR_dry_* for EGR_G–M′_ measurements because they provide a GR response integrated over a wide energy spectrum. The fitted parameters were *GR_dry_* = 99.86 nSv h^−1^ and *α* = 0.37 for the present study. The corresponding RMSE for SM estimation using the EGR_G–M′_ measurements was 6.90 vol. %. This result is consistent with the findings of Akter et al. [24] where values for *α* ranged between 0.2 and 0.3 and the RMSE value ranged between 7 and 9 vol. %. The increased uncertainty in SM predictions from EGR_G–M′_ measurements relative to reference ^40^K GR (1460 keV) measurements can be attributed to the influence of confounding factors, as discussed below.

One of the possible factors affecting the accuracy of the soil water content measurements from GR measurements is aboveground biomass [15,27]. Figure 6 shows that ^40^K GR measurements in the presence of biomass are consistently lower for a given SM. This indicates that aboveground biomass causes additional attenuation of GR signal. However, the additional attenuation of ^40^K GR signal was not clearly evident during the period with dry biomass. This is attributed to the size of hydrogen pool in fresh and dry biomass. The winter wheat canopy contains ~80% water during the growth stage, which significantly decreases as the plant matures and is reduced to 30–40% at maturity [52]. Therefore, the higher hydrogen content in fresh biomass caused greater attenuation of the ^40^K GR signal compared to dry biomass. This finding is in agreement with Baldoncini et al. [27], who collected tomato plant samples at four different maturity stages using a destructive sampling method. They estimated biomass water content through gravimetric analysis and applied a linear fit to represent biomass water content throughout the entire growing season. Based on this, they developed a biomass correction factor for ^40^K GR measurements by modeling biomass water content of the tomato plant as an equivalent water layer above the soil surface and performing Monte Carlo simulations with a progressively increasing water layer thickness. Becker et al. [15] also used the same method but they additionally considered dry biomass to calculate the biomass-equivalent water layer above the soil surface to account for the presence of maize and soybean during the measurement period. The presence of biomass attenuated the signal more strongly in case of EGR_G–M′_ measurements as compared to the ^40^K GR measurements. Consequently, excluding the period with fresh biomass from the analysis led to a substantial improvement in the SM predictions from EGR_G–M′_ (RMSE reduced from 6.90 to 4.93 vol. %) (Figure A6). This stronger effect on EGR_G–M′_ is likely due to the considerable fraction of GR with low energy measured by the MIRA detector, as also indicated by the low value of α = 0.37 as compared to α = 1.11 for ^40^K GR measurements. Based on the results presented in this study, it can be concluded that low-energy GR is more strongly attenuated by biomass as compared to high-energy GR. Both biomass water content and dry biomass are required to understand the shielding effect of biomass on EGR_G–M′_ measurements.

In addition to biomass, detector-specific characteristics, such as measured energy range and energy sensitivity, may also have contributed to the higher uncertainty in SM predictions from the EGR_G–M′_ measurements. For example, we obtained more accurate SM estimates using EGR measurements from the SARA detector (EGR_S_) compared to that of the MIRA detector. In particular, the calibrated α value shifted from 0.37 to 0.53 and the RMSE reduced from 6.90 to 4.32 vol. % when EGR_S_ measurements were used instead of EGR_G–M′_ measurements (Figure 6 and Figure A7). The more advanced SARA detector provides EGR measurements within the energy range of 500–3000 keV. Therefore, the EGR_s_ measurements of this detector were free from noise related to high-energy gamma rays (>3000 keV). Moreover, SARA provides a more reliable response in the energy range of typical natural soil radionuclides, since the system was calibrated using a ^40^K source (see Figure A1). On the other hand, the MIRA detector measured a considerable amount of low-energy gamma rays (<500 keV) and was also affected by high-energy gamma rays (>3000 keV). Thus, SM can be estimated with a 2.6 vol. % lower prediction uncertainty using EGR_S_ measurements than EGR_G–M′_ measurements.

Gamma radiation emitted from the radionuclide ^214^Bi can be used as proxy for radon. The WTC analysis revealed a correlation between ^214^Bi and EGR_G–M′_ measurements throughout the entire investigation period, which may also explain the lower SM prediction accuracy from EGR_G–M′_ measurements compared to ^40^K GR measurements. Whereas the SM estimates from ^40^K measurements are hardly affected by radon-induced GR as only one energy channel (1460 keV) was used for the analysis, and GR from both ^40^K and ^214^Bi are embedded in the EGR_G–M′_ measurements. The WTC analysis showed that the ^214^Bi signal explained part of the EGR_G–M′_ variations during wintertime. Therefore, we assume that radon-induced noise influenced EGR_G–M′_ measurements and is also partly responsible for the increased uncertainty in SM predictions. It should be noted that this radon-induced noise influenced the MIRA detector more compared to the SARA detector. As discussed earlier, the MIRA detector is more sensitive to high-energy GR (>1000 keV) where ^214^Bi emits gamma rays (1700–2200 keV, [53]). The higher magnitude of EGR_G–M′_ measurements during wintertime relative to EGR_S_ measurements (Figure 3d) also supports this finding. Consequently, the scattering of the data was more prominent for EGR_G–M′_ measurements compared to the EGR_S_ measurements during wintertime when ^214^Bi made a high contribution to EGR (see the datapoints representing 25–40 vol. % SM in Figure 6 and Figure A7).

Lastly, the higher uncertainty in the EGR_G–M′_-based model can also be partially attributed to the calibrated α parameter in comparison to the ^40^K GR-based model where the α parameter is known. In the low-energy range, the α value changes sharply with small variations in gamma-ray energy which is not the case in the high-energy range (see Figure S1 in Akter et al. [24]). For example, the α values corresponding to gamma-ray energies of 30, 40, and 50 keV are 0.37, 0.53, and 0.67, respectively, while for high-energy gamma rays (>500 keV), the α value is approximately 1.11 [39]. In the present study, the sensitivity of the α parameter also influences the RMSE of the EGR_G–M′_-based SM prediction model, as the RMSE shifts from 6.90 to 5.63 vol. % when the calibrated α parameter changes from 0.37 to 0.50.

## 4. Conclusions

In this study, we compared the accuracy of soil moisture (SM) estimates derived from environmental gamma radiation measured by a low-cost Geiger–Mueller (G–M) counter-based detector (EGR_G–M′_) and gamma radiation from the decay of ^40^K measured by a co-located gamma spectrometer. For a better understanding of confounding factors responsible for higher SM prediction error and better interpretation of EGR_G–M′_ measurements, we also analyzed EGR_S_ (comparable to EGR_G–M′_ but derived for spectral measurements) and GR from ^214^Bi (as a proxy for radon-induced noise). Wavelet coherence analysis on daily GR time series revealed that EGR_G–M′_ measurements captured distinct changes in terrestrial radiation related to SM variability during summertime. The weak correlation between these two variables in wintertime was attributed to soil processes (i.e., wet soil with limited SM variability) and the influence of radon-induced noise in GR measurements. This factor was related to higher near-surface radon concentrations, driven by reduced vertical mixing in the atmosphere as well as variable exhalation from the soil, to which EGR_G–M′_ was more sensitive. Another confounding factor, biomass hydrogen content, also affected the EGR_G–M′_ measurements. The reference ^40^K GR measurements were not influenced by radon-induced noise, and less affected by biomass compared to EGR_G–M′_ measurements. Consequently, we achieved a good fit to the data using the theoretically derived value of 1.11 for the *α* parameter in estimating SM from the ^40^K GR measurements. The resulting R^2^ was 0.88 and the RMSE for predicting SM was 3.4 vol. %. In future studies, it will be explored whether biomass corrections can further improve this accuracy. On the other hand, the higher influence of confounding factors on the EGR_G–M′_ measurements resulted in a higher prediction error of 6.9 vol. % (increase of 3.5 vol. %). To improve accuracy, radon-induced noise, detector-specific energy sensitivity, and biomass effects should be addressed. However, this can be challenging in the context of utilizing bulk EGR data from operational radioactivity monitoring networks for large-scale SM monitoring. However, an additional correction for bulk EGR data to account for GR attenuation by biomass following the approach of Baldoncini et al. [27] and Becker et al. [15] may be feasible, provided biomass hydrogen content data are available from field measurements or satellite observations. This will be explored in a follow-up study. In summary, this study enhanced our understanding of how confounding factors influence bulk EGR measurements in the context of SM estimation, and to what extent this leads to reduced prediction accuracy.

## Figures and Tables

**Figure 1 sensors-25-04453-f001:**
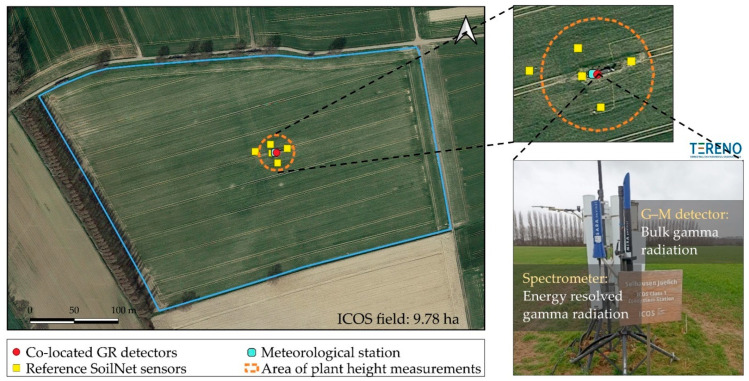
Site overview of the Selhausen TERENO/ICOS observatory. Insets show the experimental setup and the co-located gamma radiation (GR) detectors. Acronyms: TERENO—TERrestrial ENvironmental Observatories; ICOS—Integrated Carbon Observation System.

**Figure 2 sensors-25-04453-f002:**
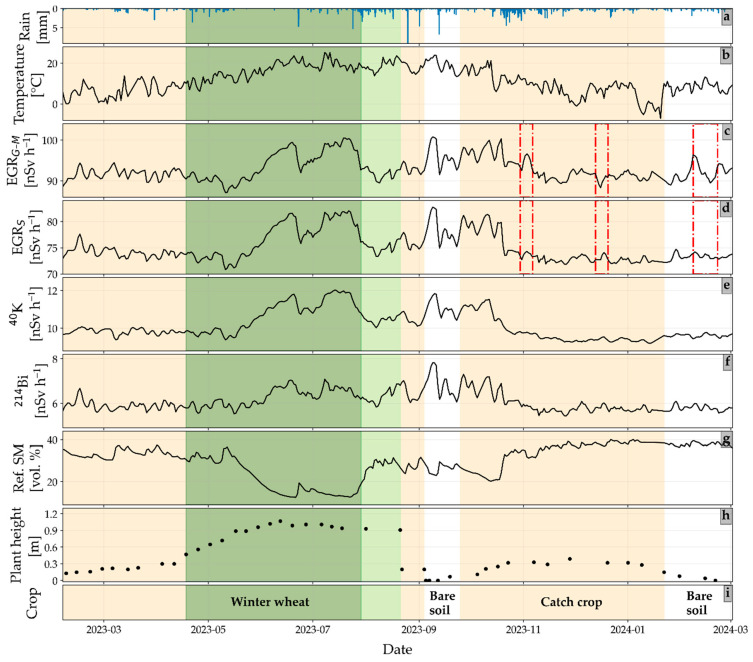
Daily time series of (**a**) precipitation, (**b**) temperature, (**c**) environmental gamma radiation measured by the MIRA detector (EGR_G–M_), (**d**) environmental gamma radiation measured by the SARA detector (EGR_S_), (**e**) gamma radiation (GR) from ^40^K measured by the SARA detector, (**f**) gamma radiation (GR) from ^214^Bi measured by the SARA detector, (**g**) reference soil moisture (Ref. SM), (**h**) plant height, and (**i**) crop rotation. All the data are shown in daily resolution except rain (10 min) and plant height (weekly/biweekly). The beige areas indicate the periods with crops, the dark green area indicates the period with high fresh biomass and the light green area indicates the period with dry biomass (after ripening, winter wheat crops lost water and dried almost completely before harvest). The vertical red lines indicate periods with discrepancies between EGR_G–M_ and EGR_S_ measurements.

**Figure 3 sensors-25-04453-f003:**
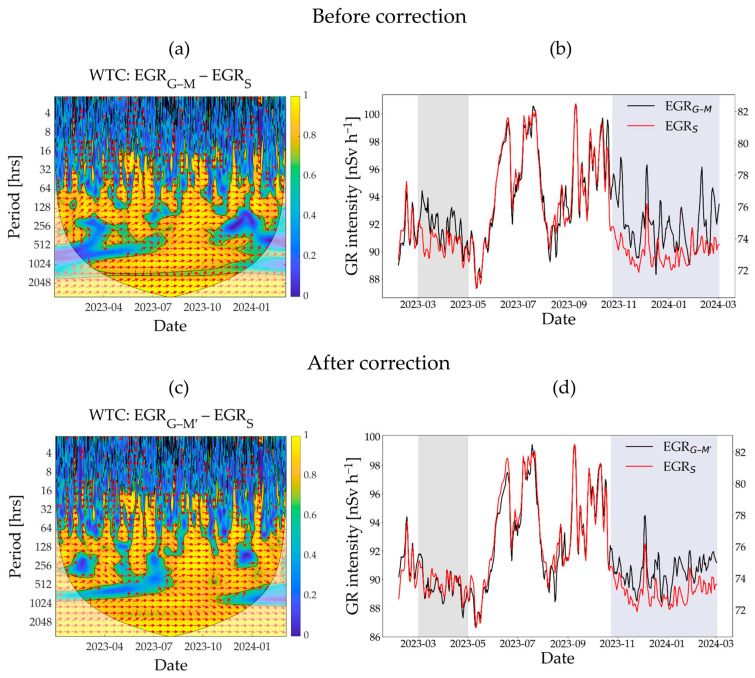
The WTC analysis results showing correlation between MIRA and SARA detectors in terms of environmental gamma radiation measurements: (**a**,**c**) show WTC plots before and after correcting MIRA measurements, respectively, across different time frequencies; (**b**,**d**) present the corresponding daily time series. WTC, wavelet transform coherence; EGR_G–M_, environmental gamma radiation measured by the MIRA detector; EGR_G–M′_, corrected environmental gamma radiation measured by the MIRA detector; EGR_S_, environmental gamma radiation measured by the SARA detector. The time windows highlighted by light gray and navy blue show the periods with discrepancies between the two types of gamma radiation detectors. In the WTC plots, the orientation of the red arrows indicates how two time series are related in time and frequency: right-pointing arrows indicate positive correlation, left-pointing arrows indicate negative correlation, straight up- or down-pointing arrows indicate a phase shift of −90° or +90°, respectively, and diagonal arrows indicate intermediate phase differences between two time series. The color spectrum shows correlation strength, with blue indicating weak and yellow indicating strong correlation.

**Figure 4 sensors-25-04453-f004:**
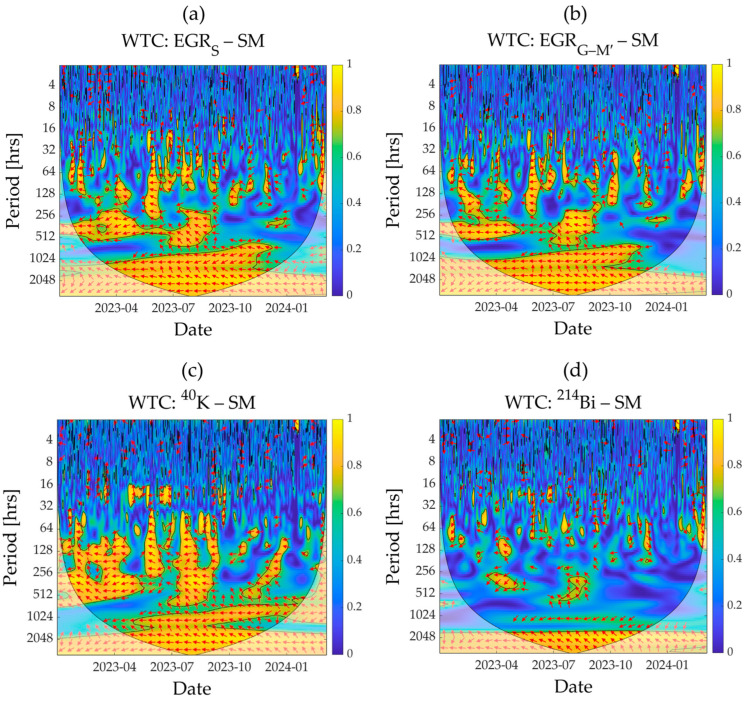
The WTC analysis results showing the correlation between reference soil moisture (SM) and different types of gamma radiation (GR) measurements obtained from both detectors: (**a**) EGR_S_, (**b**) EGR_G–M′_, (**c**) ^40^K, and (**d**) ^214^Bi. WTC, wavelet transform coherence; EGR_S_, environmental gamma radiation measured by the SARA detector; EGR_G–M′_, corrected environmental gamma radiation measured by the MIRA detector; ^40^K, GR from ^40^K radionuclide measured by the SARA detector; ^214^Bi, GR from ^214^Bi radionuclide measured by the SARA detector. The orientation of the red arrows indicates how two time series are related in time and frequency: right-pointing arrows indicate positive correlation, left-pointing arrows indicate negative correlation, straight up- or down-pointing arrows indicate a phase shift of −90° or +90°, respectively, and diagonal arrows indicate intermediate phase differences between two time series. The color spectrum shows correlation strength, with blue indicating weak and yellow indicating strong correlation.

**Figure 5 sensors-25-04453-f005:**
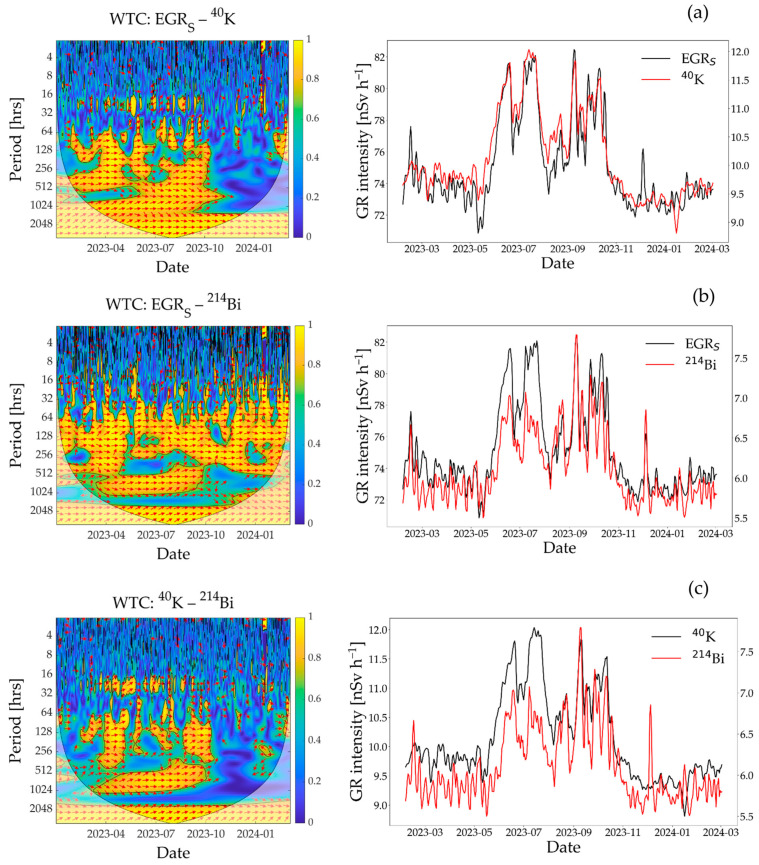
The results of the WTC analysis showing correlation between different types of gamma radiation (GR) measurements obtained from the SARA detector across different time frequencies. Each subplot includes the WTC plot (left) and corresponding daily time series (right) for the following pairs: (**a**) EGR_S_ and ^40^K, (**b**) EGR_S_ and ^214^Bi, and (**c**) ^40^K and ^214^Bi measurements. WTC, wavelet transform coherence; EGR_S_, environmental gamma radiation obtained from the SARA detector; ^40^K, GR from ^40^K radionuclide measured by the SARA detector; ^214^Bi, GR from ^214^Bi radionuclide measured by the SARA detector. The orientation of the red arrows indicates how two time series are related in time and frequency: right-pointing arrows indicate positive correlation, left-pointing arrows indicate negative correlation, straight up- or down-pointing arrows indicate a phase shift of −90° or +90°, respectively, and diagonal arrows indicate intermediate phase differences between two time series. The color spectrum shows correlation strength, with blue indicating weak and yellow indicating strong correlation.

**Figure 6 sensors-25-04453-f006:**
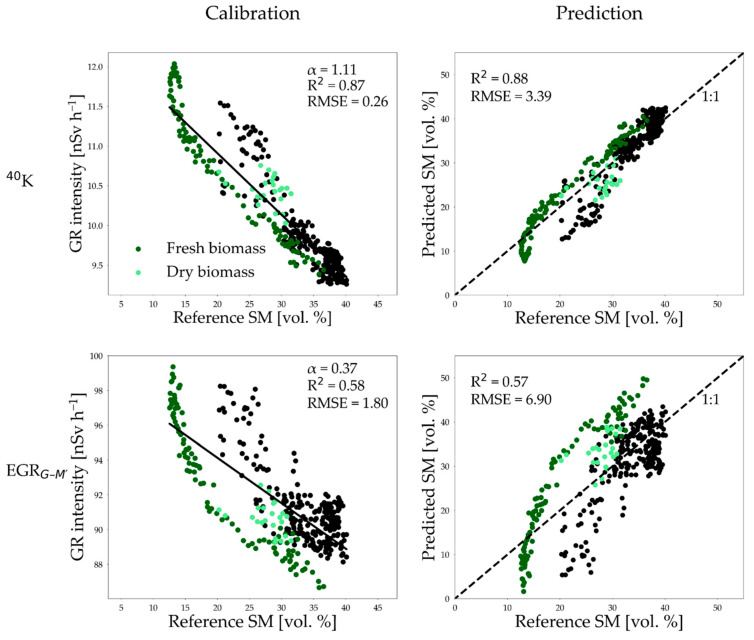
Comparison of the accuracy of SM estimation obtained from EGR_G–M′_ and ^40^K measurements. SM, soil moisture; EGR_G–M′_, corrected environmental gamma radiation measured by the MIRA detector; ^40^K, gamma radiation from ^40^K radionuclide measured by the SARA detector; α, the effective ratio of gamma-ray mass attenuation coefficients for water and solid phases. The dark green datapoints represent the period with fresh biomass and the light green datapoints represent the period with dry biomass.

**Table 1 sensors-25-04453-t001:** Summary statistics of daily gamma radiation (GR) time series.

Measurements	Mean [nSv h^−1^]	Maximum [nSv h^−1^]	Minimum [nSv h^−1^]	Coefficient of Variation
EGR_G–M_ ^1^	92.98	100.76	87.12	0.032
EGR_G-M′_ ^2^	91.71	99.49	86.66	0.031
EGR_S_ ^3^	75.18	82.76	70.84	0.038
GR from ^40^K ^4^	10.17	12.04	9.20	0.073
GR from ^214^Bi ^5^	6.11	7.82	5.43	0.075

^1^ EGR_G–M_, environmental gamma radiation measured by the MIRA detector; ^2^ EGR_G–M′_, corrected environmental gamma radiation measured by the MIRA detector; ^3^ EGR_S_, environmental gamma radiation measured by the SARA detector; ^4^ GR from ^40^K, gamma radiation from ^40^K radionuclide measured by the SARA detector; ^5^ GR from ^214^Bi, gamma radiation from ^214^Bi radionuclide measured by the SARA detector.

## Data Availability

The complete data set employed in this study has been made publicly available on the DETECT-Geonetwork platform (https://detect-z03.geoinformation.net/geonetwork/srv/eng/catalog.search#/search (accessed on 2 December 2024)). Furthermore, climate data used in this study is available on the TERENO (Terrestrial Environmental Observatories) data portal (https://ddp.tereno.net/ddp/ (accessed on 14 October 2024)).

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
