# Peer review of "Comparing the Accuracy of Soil Moisture Estimates Derived from Bulk and Energy-Resolved Gamma Radiation Measurements"

_sensors, 2025, doi:10.3390/s25144453_

Round 1
Reviewer 1 Report
Comments and Suggestions for Authors
The manuscript entitled Comparing the accuracy of soil moisture estimates derived from bulk gamma radiation and energy-resolved gamma radiation measurements by Akler etal generally has a good scientific flaw. However, several points need to be considered to be revised. my decision is Major revision. the detail review are below
Abstract
Please make it shorter and sharper. In the abstract, you should mention briefly the background, novelty claim, method, result, and further study.
method
- please add the detail of calibration of detector?
- in GM counter, how to distinguish energy peak? as we know in the nature have many radionuclide especially from NORM?
- which energy the author used for scintillation detector?
- Line 206 - 208, are you also try to used modelling approace? such montecarlo?
-how the author can be correcting the energy from cosmic ray for GM counter?
Discusion
Why did the author choose the dose equivalent rate over the absorbed dose rate? in my opinion the absorbed dose rate is more realistic for the unit
Figure 3, 4 and 5 need to brushed up for better resolution
Author Response
Dear Reviewer,
We appreciate the time and effort that you have invested in evaluating our manuscript. We have tried our best to address the major and minor comments given by you. Please find below a point-by-point reply to your comments. We think that the manuscript benefited considerably from the reviewing and editing process.
Best Regards,
Sonia Akter (on behalf of all co-authors)
Reviewer 1
The manuscript entitled Comparing the accuracy of soil moisture estimates derived from bulk gamma radiation and energy-resolved gamma radiation measurements by Akter et al. generally has a good scientific flow. However, several points need to be considered to be revised. My decision is Major revision. The detailed review is given below.
Abstract.
Please make it shorter and sharper. In the abstract, you should mention briefly the background, novelty claim, method, result, and further study.
REPLY: We thank the reviewer for dedicating time and effort to review our work. Following the comments of the reviewer, we have improved the abstract.
Method.
- please add the detail of calibration of detector?
REPLY: Added in lines 194–202 and 215–219.
- in GM counter, how to distinguish energy peak? as we know in the nature have many radionuclide especially from NORM?
REPLY: The GM counter operates on a non-spectroscopic principle, differing fundamentally from spectrometric detectors. It detects and counts radiation events but provides no information on their energy. As a result, GM counters can measure the presence of radiation but are unable to determine the specific radionuclide responsible, highlighting a fundamental limitation of these devices. This information was already provided in lines 96–103 and 209–211. However, we have further improved lines 209–211 for better clarity.
- which energy the authors used for scintillation detector?
REPLY: For the scintillation detector (SARA spectrometer for the current study), measurements such as bulk EGR (EGRS), GR from 40K and 214Bi were used. For the 40K GR, only one energy channel at 1460 keV was used (mentioned in line 135). For the 214Bi GR, characteristic gamma photopeaks or gamma lines throughout the measured energy range of SARA (30–3000 keV) were used. We improved lines 189–191 to emphasize this. For calculating EGRS, all gamma lines of the spectrum emitted from all ambient radionuclides in the given energy range of the spectrometer were considered. We have improved lines 192–194 for better clarity.
- Line 206 - 208, do you also try to use modelling approach? such Monte Carlo?
REPLY: No, we used equation 4 from Grasty (1987) to calculate the horizontal footprint and equation 5 from Grasty (1997) to calculate the vertical footprint of the GR sensor for different gamma-ray energies. Detailed information can be found in the Supplemental material of Akter et al. (2024). Our findings were in agreement with Baldoncini et al. (2018), who used Monte Carlo simulation for calculating the GR sensor footprint. All this information has been documented in previous studies, which is why we have only briefly mentioned the topic of sensor footprint in the current study.
-how the author can be correcting the energy from cosmic ray for GM counter?
REPLY: Primary cosmic rays coming from outer space interact with the atmosphere of the earth creating secondary cosmic radiation (SCR). The measurements of GM counter also include the contribution of this component. Therefore, the contribution of SCR needs to be subtracted from the EGR measurements. The long-term fraction of SCR is assumed to be constant for the experimental site, but short-term variations in SCR can occur due to air pressure and incoming neutron intensity. This can lead to variations in EGR measurements made with a GM counter. Therefore, it is sufficient to correct the EGR measurements for short-term SCR variability for the purpose of our study. The correction of EGR for SCR contribution is well documented in Akter et al. (2024). In the current study, we added lines 249–253 and 260–261 for further clarification in the methodology.
Discussion
Why did the author choose the dose equivalent rate over the absorbed dose rate? in my opinion the absorbed dose rate is more realistic for the unit.
REPLY: We preferred to use the dose equivalent rate over the absorbed dose rate because the absorbed dose rate implies the measurements of physical energy absorbed by air or material while the dose equivalent rate reflects the estimated biological risk of gamma radiation. In radiation dosimetry, the dose equivalent rate is generally used instead of the absorbed dose rate as the biological impact is the more practical measure for safety decisions in the environment. This study aims to improve the interpretation of bulk gamma radiation measurements from radiation monitoring networks to enhance the accuracy of soil moisture estimation. Since such measurements are commonly expressed in terms of dose equivalent rate, this metric is adopted throughout the manuscript.
Figure 3, 4 and 5 need to brushed up for better resolution
REPLY: We improved the resolution of figures 3, 4 and 5.
References
Akter, S., Huisman, J. A., & Bogena, H. R. (2024). Estimating soil moisture from environmental gamma radiation monitoring data. Vadose Zone Journal, e20384. https://doi.org/10.1002/vzj2.20384
Baldoncini, M., Albéri, M., Bottardi, C., Chiarelli, E., Raptis, K. G., Strati, V., & Mantovani, F. (2018). Investigating the potentialities of Monte Carlo simulation for assessing soil water content via proximal gamma-ray spectroscopy. Journal of Environmental Radioactivity, 192, 105–116. https://doi.org/10.1016/j.jenvrad.2018.06.001
Grasty, R. L. (1987). The design, construction, and application of airborne gamma-ray spectrometer calibration pads – Thailand. Ottawa, Canada: Geological Survey of Canada.
Grasty, R. L. (1997). Radon emanation and soil moisture effects on airborne gamma-ray measurements. Geophysics, 62, 1379–1385. https://doi.org/10.1190/1.1444242
Reviewer 2 Report
Comments and Suggestions for Authors
Dear Author/s,
The submitted manuscript is well-written and discussing about the possibility of estimating soil moisture by low-cost, permanently installed Geiger–Mueller (G–M) bulk gamma detectors and measuring the 40K line at 1460 keV. It is timely, being appropriate to the operational potential of environmental gamma radiation (EGR) monitoring networks for the purposes of hydrology and agriculture, a subject of mounting importance inasmuch as such networks are being established for use worldwide.
The paper is generally well written, methodologically correct, and contributes useful new information, especially for the confounding factors of the EGR-based soil moisture estimations, however, the manuscript lacks of some fundamental discussion regarding biomass limitation factor. As the impact of the aboveground biomass is proved and acknowledged to be one of the main limitation, the paper does not attempt to propose an operational workflow/correction model. For the applicability in practice, the authors (I know it might get time and efforts, even a short discussion or theoretical discussion is enough) should, (1) should suggest or discussed about possible abouth a validate simple biomass correction scheme, perhaps leveraging published models or their own plant height/water content data; (2) then please in conclusion state this limitation and give a suggestion for more investigation for future work.
Other area that needs your attention is about the influencing factors and the sources of uncertainty (instrumental, environmental, model, reference SM), the paper could benefit a short discussion about:
-
How the α parameter for EGR calibration could have influence or cause some what uncertainty on results?
-
what is the expected results and outcomes of SM reference (SoilNet) during periods of sensor removal or soil disturbance?
-
did authors run any cross-validations performed to test prediction stability?
I would suggest the manuscript for publication after applying mentioned correction.
Best regards,
Author Response
Dear Reviewer,
We appreciate the time and effort that you have invested in evaluating our manuscript. We have tried our best to address the major and minor comments given by you. Please find below a point-by-point reply to your comments. We think that the manuscript benefited considerably from the reviewing and editing process.
Best Regards,
Sonia Akter (on behalf of all co-authors)
Reviewer 2
Dear Author/s,
The submitted manuscript is well-written and discussing about the possibility of estimating soil moisture by low-cost, permanently installed Geiger–Mueller (G–M) bulk gamma detectors and measuring the 40K line at 1460 keV. It is timely, being appropriate to the operational potential of environmental gamma radiation (EGR) monitoring networks for the purposes of hydrology and agriculture, a subject of mounting importance in as much as such networks are being established for use worldwide.
The paper is generally well written, methodologically correct, and contributes useful new information, especially for the confounding factors of the EGR-based soil moisture estimations, however, the manuscript lacks of some fundamental discussion regarding biomass limitation factor. As the impact of the aboveground biomass is proved and acknowledged to be one of the main limitation, the paper does not attempt to propose an operational workflow/correction model. For the applicability in practice, the authors (I know it might get time and efforts, even a short discussion or theoretical discussion is enough) should, (1) should suggest or discussed about possible abouth a validate simple biomass correction scheme, perhaps leveraging published models or their own plant height/water content data; (2) then please in conclusion state this limitation and give a suggestion for more investigation for future work.
REPLY: We thank the reviewer for his appreciation of our work. We also thank the reviewer for this comment on including a method on biomass correction. However, we feel that the topic of better understanding the influence of confounding factors on EGR measurements warrants a paper of its own, whereas the inclusion of a method for biomass correction would be beyond the scope of this paper. However, considering the suggestion of the reviewer, we added more information on available biomass correction methods from previous studies in lines 562–565 and 567–568. In addition, we mentioned the need for further investigations in a follow-up study in the Conclusions (line 653).
Other area that needs your attention is about the influencing factors and the sources of uncertainty (instrumental, environmental, model, reference SM), the paper could benefit a short discussion about:
- How the α parameter for EGR calibration could have influence or cause some what uncertainty on results?
REPLY: We thank the reviewer for this insightful comment. We added this discussion as a separate paragraph in the manuscript (lines 614–623).
- what is the expected results and outcomes of SM reference (SoilNet) during periods of sensor removal or soil disturbance?
REPLY: In our view, the reference SM measurements during the periods of sensor removal or soil disturbance had minimal impact on the model uncertainty. The gap-filled reference SM data - based on the SM variability trend from an intact SoilNet station (see lines 266–270 for details) - appeared reliable, as illustrated in Figure R1. Furthermore, excluding the one-month gap-filled data did not lead to any improvement in model performance. After ploughing and planting cover crops, four SoilNet stations were reinstalled in the experimental field. To minimize model uncertainty, one week of data following this soil disturbance event was excluded from the SM prediction model (see lines 346–348).
Figure R1. Reference soil moisture (SM) measurements. L1, location 1.
- did authors run any cross-validations performed to test prediction stability?
REPLY: Thank you for this insightful comment. We agree that a cross-validation technique helps to prevent model overfitting. However, we think that better understanding of the GM detector signal or applying corrections is necessary prior to perform cross-validation. Therefore, we focused on improving the understanding of the GM detector signal and assessing the influence of different confounding factors (i.e., environmental and instrumental) over a one-year period. Once the signal is better characterized and appropriate corrections are applied, cross-validation or transferring the results to different sites can be pursued in future studies, e.g. by incorporating biomass correction (see comment above) to evaluate prediction stability.
References
Akter, S., Huisman, J. A., & Bogena, H. R. (2024). Estimating soil moisture from environmental gamma radiation monitoring data. Vadose Zone Journal, e20384. https://doi.org/10.1002/vzj2.20384
Round 2
Reviewer 1 Report
Comments and Suggestions for Authors
I think the paper can be published in the current form
Reviewer 2 Report
Comments and Suggestions for Authors
Dear Author/s,
Thanks for efforts and appling the comments. Paper has been improved both in quality and quantity.
Regards,